# Mutational processes of distinct POLE exonuclease domain mutants drive an enrichment of a specific TP53 mutation in colorectal cancer

Hu Fang[1], Jayne A. Barbour[1,2], Rebecca C. Poulos[3], Riku Katainen[4,5], Lauri A. Aaltonen[4,5], Jason W. H. Wong[1,2]*

1 School of Biomedical Sciences, Li Ka Shing Faculty of Medicine, The University of Hong Kong, Hong Kong Special Administrative Region, 2 Prince of Wales Clinical School, UNSW Medicine, UNSW Sydney, New South Wales, Australia, 3 Children's Medical Research Institute, Faculty of Medicine and Health, The University of Sydney, Westmead, New South Wales, Australia, 4 Applied Tumor Genomics Research Program, Faculty of Medicine, University of Helsinki, Helsinki, Finland, 5 Department of Medical and Clinical Genetics, Medicum, University of Helsinki, Helsinki, Finland

* jwhwong@hku.hk

**Data Availability Statement:** Data used from this study can be obtained from the NCI Genomic Data Commons Data Portal (https://portal.gdc.cancer.

## Abstract

Cancer genomes with mutations in the exonuclease domain of Polymerase Epsilon (POLE) present with an extraordinarily high somatic mutation burden. *In vitro* studies have shown that distinct POLE mutants exhibit different polymerase activity. Yet, genome-wide mutation patterns and driver mutation formation arising from different POLE mutants remains unclear. Here, we curated somatic mutation calls from 7,345 colorectal cancer samples from published studies and publicly available databases. These include 44 POLE mutant samples including 9 with whole genome sequencing data available. The POLE mutant samples were categorized based on the specific POLE mutation present. Mutation spectrum, associations of somatic mutations with epigenomics features and co-occurrence with specific driver mutations were examined across different POLE mutants. We found that different POLE mutants exhibit distinct mutation spectrum with significantly higher relative frequency of C>T mutations in POLE V411L mutants. Our analysis showed that this increase frequency in C>T mutations is not dependent on DNA methylation and not associated with other genomic features and is thus specifically due to DNA sequence context alone. Notably, we found strong association of the TP53 R213* mutation specifically with POLE P286R mutants. This truncation mutation occurs within the TT[C>T]GA context. For C>T mutations, this sequence context is significantly more likely to be mutated in POLE P286R mutants compared with other POLE exonuclease domain mutants. This study refines our understanding of DNA polymerase fidelity and underscores genome-wide mutation spectrum and specific cancer driver mutation formation observed in POLE mutant cancers.

gov/) and the cBioPortal for Cancer Genomics (https://www.cbioportal.org/). These data can be retrieved from the portals by using the sample IDs listed in S2 Table.

**Funding:** This project is supported by a Project Grant from the National Health and Medical Research Council (NHMRC), Australia (APP1119932) to J.W.H.W. R.C.P. is supported by an NHMRC Early Career Fellowship (APP1138536). R.K. is supported by the Juhani Aho Foundation for Medical Research, the Ida Montin Foundation and the Instrumentarium Science Foundation. R.K. and L.A.A. are supported by the Academy of Finland (Finnish Center of Excellence Program 2018-2025, 312041). The funders had no role in study design, data collection and analysis, decision to publish, or preparation of the manuscript.

**Competing interests:** The authors have declared that no competing interests exist.

## Author summary

Cancer arises through the accumulation of somatic mutations. The way that these somatic mutations form can vary greatly in different cancers. One of the most mutagenic processes that have been identified is caused by mutations within a replicative DNA polymerase known as Polymerase Epsilon (POLE). Cancers with such mutations present with hundreds of thousands of somatic mutations in their genome. Previous cancer genomics studies have identified a number of mutation hotspots in POLE, however how these different POLE mutants behave in affecting mutation distribution has not been studied. Here, we describe the genome-wide mutation profiles of distinct POLE mutant cancers. We find that different mutants indeed result in different mutation profiles and that this can be explained by the different fidelities of these mutants in replicating specific DNA sequences. Significantly, these differences have important implications in cancer formation as we found that a POLE mutation is strongly associated with a specific truncation of the TP53 cancer driver gene. This study furthers our understanding of the POLE mutagenic process in cancer and provide important insights into carcinogenesis in cancers with such mutations.

## Introduction

*POLE* encodes the catalytic subunit of DNA Polymerase Epsilon, which is responsible for DNA fidelity during the process of eukaryotic nuclear genome replication [1]. Functional POLE mutations have been identified in less than 1% of all cancer genomes but these genomes are characterized by exceptionally high tumor mutation burden [2]. Somatic mutations of POLE exonuclease domain are frequently enriched in brain, uterine and colorectal cancer [3], and patients with POLE dysfunction usually have significantly better prognosis and require less intensive treatment [4].

The POLE mutational process shapes the cancer genome into a unique mutational signature with high proportions of C>A mutations at TCT contexts, C>T mutations at TCG contexts and T>G mutations at TTT contexts, which is known as COSMIC signature 10 [5]. Several driver mutations have been identified in the *POLE* exonuclease domain (codons 268–471) [6], the most frequent being P286R and V411L [2]. The crystal structure of the yeast orthologue has shown that P301R (P286R in Human) could change the exonuclease domain, with R301 pointing towards the exonuclease site, leading to polymerase hyperactivity and increased capacity to extend mismatches by interfering with DNA binding to the exonuclease site [7,8]. By contrast, residue V411 lies a distance away from the binding site and does not interact with the DNA sequence directly [9]. In endometrial cancer, it has been shown that V411L and P286R display different signature fraction with V411L characterized by relative higher fraction of C>T mutations in endometrial cancer [10]. Data showing the mutation spectrum of individual POLE mutants supports differences in the way mutants generate somatic mutations [11] but these differences have not yet been quantified.

Mutations are distributed unevenly across the cancer genome and mutation rates across genomic regions are highly heterogeneous [12] due to genomic and epigenetic features including cytosine methylation [13], replication timing [14], tri-nucleotide/penta-nucleotide context composition [5], transcription factor binding, chromatin organization [15], gene expression levels [16], orientation of the DNA minor groove around nucleosomes [17], CTCF binding [18] and gene body features such as introns and exons [19]. As POLE mutant cancers are

usually hypermutated and individual mutants might lead to distinct mutator phenotypes, the precise mechanisms of mutagenesis may be revealed by investigating whether they show disparity in mutational spectrum and distribution across genomic regions.

In this study, we first characterized 53 whole genomes of colorectal cancer, which harbor different *POLE* exonuclease domain somatic mutations (n = 9) or are POLE wild-type (n = 44). The mutational spectrum of the different POLE mutants was compared and validated in a large cohort of 7,345 colorectal cancer samples from additional whole exome/target capture sequencing data. We also studied the association between cytosine methylation and mutation burden, and examined genome-wide mutation profiles across a range of genomics features. Finally, combining these datasets, we sought to identify associations between specific POLE mutants and the formation of driver mutation hotspots in colorectal cancer.

## Results

### Profile of mutation signatures in different POLE-mutant colorectal cancers

As a first step, a collection of 53 colorectal cancer whole genomes from The Cancer Genome Atlas were analyzed, in which 44 are POLE wild-type and microsatellite stable, and the remaining nine carried non-synonymous somatic mutations in the POLE exonuclease domain (**S1 Table**, all other mutations have been listed in **S10 Table**). We clustered these samples based on the proportion of 96 tri-nucleotide contexts and obtained four distinct groups (**Fig 1A**). The nine POLE mutants were clustered into three subgroups, which are represented as P286R (n = 3), V411L (n = 3) and Other-Exo (n = 3) (sample with other mutations in the POLE exonuclease domain, including P286H, S297F and F367S). Samples within same subgroups have very high cosine similarity and mutational signature profile (**S1A and S1B Fig**). In line with previous reports [11], all of the POLE mutants showed a high proportion of C>A and T>G mutation in TCT and TTT tri-nucleotide contexts, which resembles COSMIC signature 10 (**Fig 1B and S2 Fig**). When examining genome-wide C>T mutations, we observed a higher proportion of C to T mutations in POLE V411L mutants accounting for 33.7% of all substitutions, while there were 16.0% and 23.3% in P286R and Other-Exo mutants respectively (P286R vs V411L, P<0.001, Chi-squared test, **Fig 1C**). The differential mutation spectrum clustering of P286R from V411L mutants was also evident in an additional 32 POLE mutant colorectal cancer samples with WGS, WXS and target capture sequencing data (**S2 and S4A Figs**), confirming the differences observed in the WGS samples.

We then computed proportions of C>T mutations in different penta-nucleotide contexts to investigate differential enrichment of these mutations (**Fig 1D and S5 Fig**). All three types of POLE mutants display enriched C>T mutations in CpG contexts, with the proportion significantly higher in V411L (52.14%), compared with P286R (41.17%) and other POLE mutants (45.67%) (p < 0.0001, Chi-square test, **S4B Fig**). We also explored the penta-nucleotide context enrichment of C>A and T>G mutations, but did not find substantial differences in the frequency of these mutations in different mutants (**S4C and S4D Fig**). Based on this analysis, we can conclude that there are differences in the mutation spectra between the POLE mutants which can be largely attributable to different frequencies of C>T and C>A mutations and the relative frequency of C>T mutations at CpG dinucleotides.

### Differential mutation load of C>T mutations in POLE mutants is independent of cytosine methylation

We and others have previously reported that C>T mutation load at CpG dinucleotides in many cancer types including POLE mutant cancers show strong positive correlation with

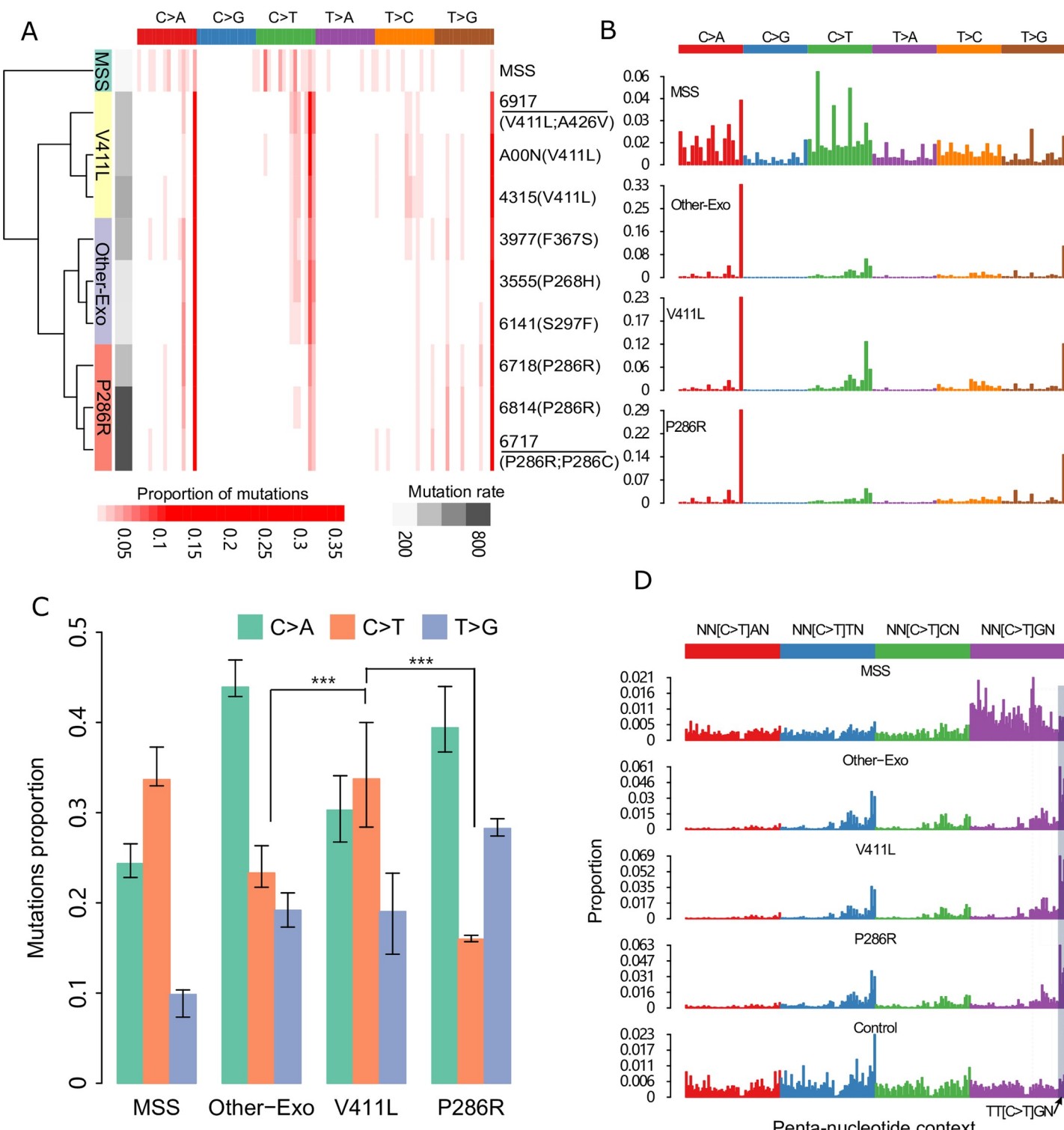

**Fig 1. Mutational spectrum of distinct colorectal cancer POLE mutants.** (**A**) Hierarchical clustered heatmap of the frequency of 96 types of mutational contexts within each WGS POLE mutant ranging from light red (0%) to dark red (35% of all mutations). Four groups "MSS (microsatellite stable), V411L, P286R and Other-Exo" were labeled on the far left panel, and total mutation burden was indicated in the right bottom. The MSS spectrum is averaged across the 44 TCGA MSS POLE wild-type WGS samples while the TCGA samples ID for each POLE mutant is shown. Mutants with multiple variants are underlined. (**B**) Mutational spectrum of four POLE-mutant groups based on 96 mutational contexts, with mutation type indicated on the top panel. (**C**) Proportion of C>A, C>T and T>G mutation in four POLE mutant groups. The significance was calculated by paired Chi-squared test. Error bars represent +/- 2 SE. (**D**) Profile of C>T mutations in penta-nucleotide contexts, with genome-wide frequency of each penta-nucleotide indicated at bottom. The detail of context information is indicated in S8 Table. *** denotes $P < 0.001$.

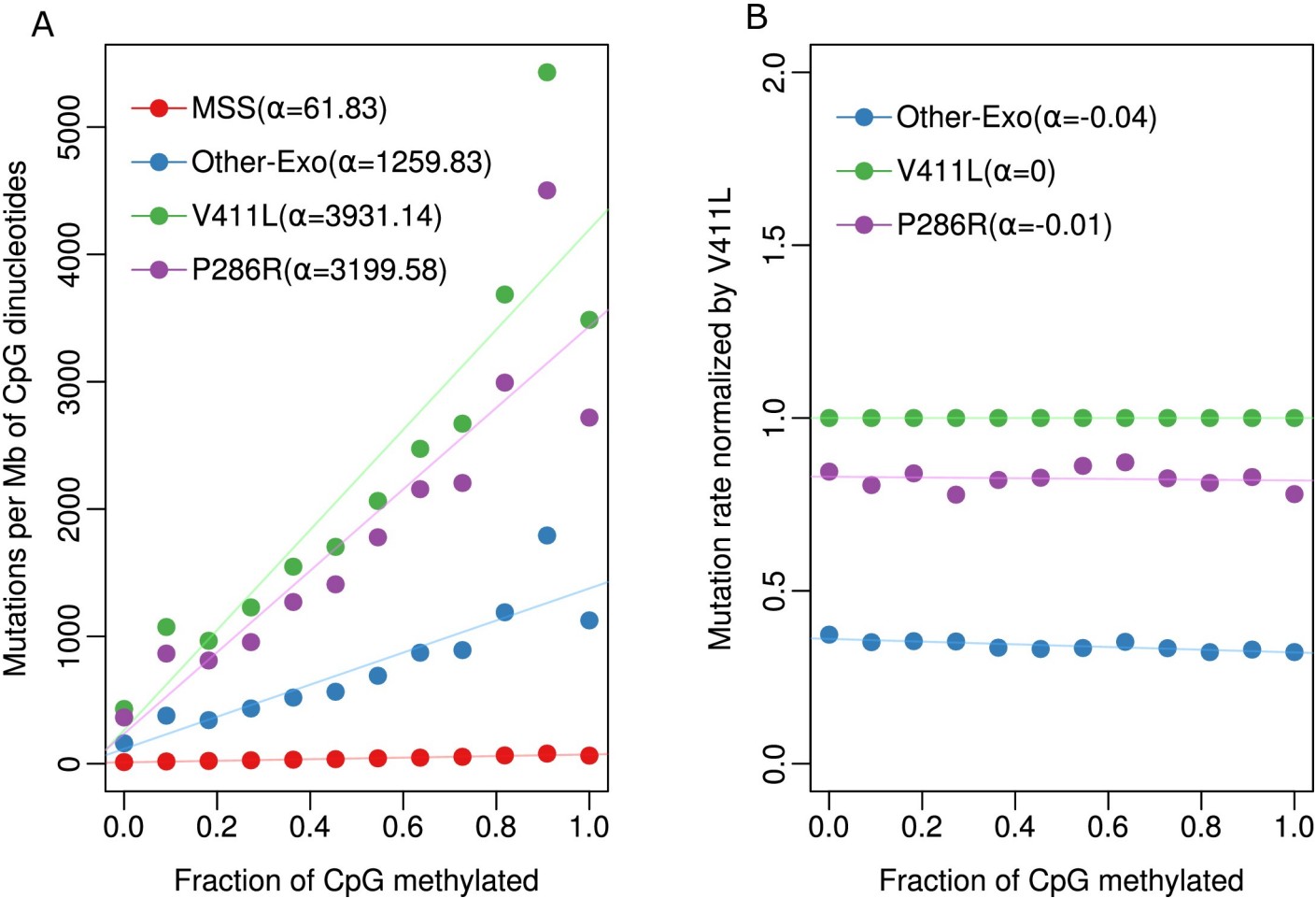

**Fig 2. Association of methylation and mutation in different POLE mutants.** (**A**) Correlation between mutations per megabase (Mb) at CpG dinucleotides and fractions of CpGs methylated across different POLE mutants and microsatellite stable (MSS) samples. (**B**) Mutation burden of each mutant was normalized by the mutation rate of V411L in each methylation level.

5-methylcytosine (5mC) level [20–22]. To investigate whether the increased C>T mutation load observed in the V411L mutants compared with P286R mutants is due to differential dependence on CpG methylation we sought to compare the relationship between 5mC level and C>T mutation frequency in the different POLE mutants. Methylation levels at CpG dinucleotides from normal sigmoid colon whole genome bisulfite sequencing data was correlated with mutational burden across the colorectal cancer genome. In all three types of POLE mutants, including POLE wild-type MSS samples, the mutation burden increased significantly with methylation levels (**Fig 2A**). To investigate whether differences in the CpG mutation load between the different POLE mutants is dependent on methylation level, mutation burden within each bin of methylation level for the different POLE mutants were normalized against V411L. We found that the slope of the normalized mutation burden does not substantially deviate from zero across increasing levels of methylation (**Fig 2B**). This finding suggests that, while mutation burden at CpG sites are dependent on 5mC levels, the relative level of dependence is the same in the different POLE mutants, thus the increased C>T mutation load in V411L compared with the other POLE mutants is likely due to sequence context alone.

## Pentanucleotide sequence context accounts for non-linear relationship between C>T load and CpG methylation level in POLE mutants

Finally, in all POLE mutants, mutation burden peaks when the level of CpG methylation measured is between 90% and 100%, but decreases when the level of CpG methylation level is equal 100%. We examined whether sequencing coverage, replication timing or repeat sequences in different methylation levels contributes to this change, but found that they were not associated with this observation (S6A–S6D Fig). We then tested the composition of penta-nucleotide contexts at different levels of methylation, since C>T mutations are also enriched in specific penta-nucleotide context as discussed above (S4B Fig). We found that there are more CpGs in the TTCGN context in the 90–100% CpG methylation bin compared with the 100% CpG methylation bin, accounting for 8.67% and 5.62% of CpGs respectively (p<0.001, Chi-square test, Fig 3A). Following normalization for penta-context composition across the different bins, the mutation rate at the 90–100% bin decreased by 17.6% (Other-Exo), 18.04% (V411L) and 20.02% (P286R) POLE mutants respectively, making the mutation rate in this bin more similar to that of the 100% methylated CpG sites (Fig 3B–3D). This finding again demonstrates that different preferences for penta-nucleotide context within POLE mutants can account for differences in the observed mutational patterns.

## Distinct POLE mutants show similar genome-wide mutational patterns

Having shown that sequence context plays a major role in the observed mutation spectra of different POLE mutants, we further sought to determine whether there are differences in mutation patterns across different epigenomic features across the genome.

Characterization of mutations in CTCF binding sites: CCCTC-binding factor (CTCF) is a transcription factor and plays an essential role in constructing three-dimensional genome organization. Somatic mutations in CTCF binding sites of the CTCF-cohesin complex (CBS) are widely observed in cancer genomes [23]. Samples with *POLE* mutations displayed lower mutation frequencies at, and adjacent to, CBS when compared with flanking regions [24], but the mutation rate of distinct POLE mutants has not been examined. We calculated mutation counts at each position within 1000 nucleotides from the CTCF motif center and we identified a distinct pattern whereby mutation burden was significantly decreased in all the three mutants (Fig 4A). For each mutant, mutation load starts to decline approximately 110 nucleotides from the CTCF motif center, and then presents a significant lower mutation frequency than expected by chance within the center of the CTCF motif, especially at the central cytosine nucleotide (P<0.001, paired Wilcoxon signed-rank test, S7A and S7B Fig). We characterized the mutation signature within this ±110 nucleotide region and we observed a similar mutational pattern with genome-wide signature (S7C Fig), suggesting that at least some of the CBS sites examined are still under the POLE mutation process.

Mutation density around the transcription start site: We also investigated mutation density around the transcription start site (TSS) in different POLE mutants. The DNA sequence around the TSS can show distinct mutation patterns, as active promoters are occupied with transcription factors, which may inhibit DNA repair access or activity [25,26]. We examined mutation profiles of C>T, C>A and T>G mutations around TSSs for each POLE mutant. Notably, before normalization, T>G mutations were substantially decreased at the TSS (S7D Fig). However, following normalization for trinucleotide sequence context, this was no longer evident, and we only observed substantial decrease in C>T mutations close to the TSS likely due to reduced DNA methylation at many gene promoters (Fig 4B).

Exonic regions show decreased mutation burden in POLE mutants: Increased mismatch repair (MMR) activity at exons compared with introns has been shown to result in a significant

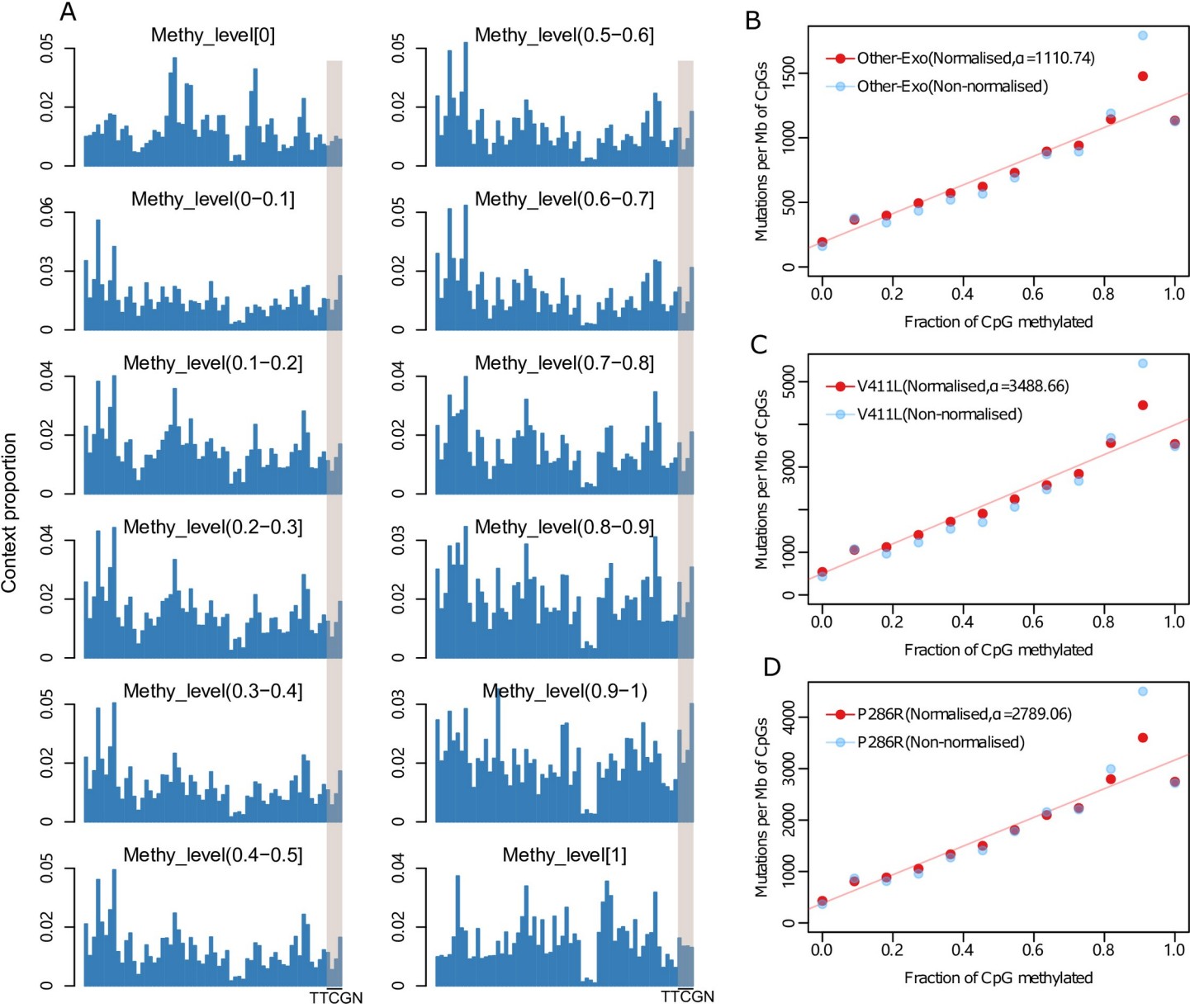

**Fig 3. Sequence context in different methylation bins. (A)** Proportion of each "NNCGN" penta-nucleotide context in different methylation level, with "TTCGN" shadowed. The detail of context information is indicated in S8 Table. **(B-D)** Correlation of methylation and mutation burden after normalization of penta-nucleotide context composition, with non-normalized data indicated in light blue.

decrease in exonic mutation rate in MMR proficient POLE mutants [19]. We investigated mutation patterns of exonic and intronic regions in different POLE mutants (**Fig 4C**). All three kinds of POLE mutants showed decreased mutation rates in exonic region. Particularly in P286R mutants, the average mutation burden in the middle of intronic regions is approximately double the count in the middle of exonic regions (260 vs 528 Mut/Mb, **S7E Fig**).

POLE mutants present mutational strand asymmetries:

Since the exonuclease domain of POLE is responsible for proofreading during synthesis of the DNA leading strand, mutations caused by deficiency of the domain should show very strong strand asymmetries [27]. We identified this phenomenon in all distinct POLE mutants,

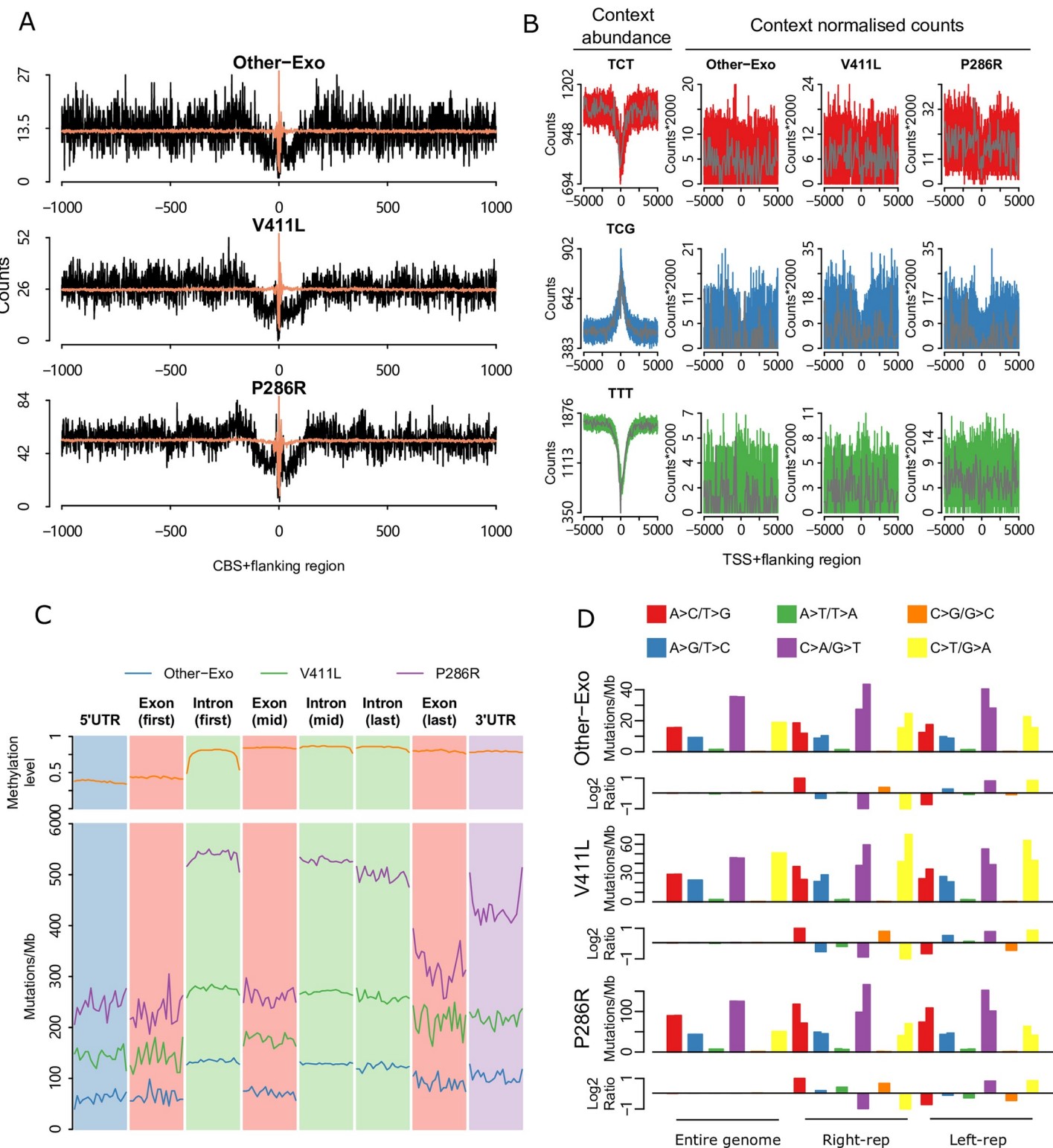

**Fig 4. Genome-wide mutational patterns of distinct POLE mutants.** (**A**) Somatic substitutions at CBSs with a flanking sequence of 1 kilo bp in different POLE mutants. The expected mutation was indicated in light red colour. (**B**) Mutation profile around transcription start sites in different mutants. Three primary mutation types C>A (red), C>T (blue) and T>G (green) in specific context were showed. Mutation counts were normalized by the number of corresponding context and the abundance of each context was displayed in the far left panel, together with mutation data in 100 bp bins (grey) is shown. (**C**) Profile of mutation burden across different part of genes in different mutants. Each part of gene was divided into 20 bins and mutation burden was calculated separately. Methylation level of each part was showed in the top panel. (**D**) Mutational strand asymmetry associated with replication in different mutants. Lower panel of each mutant shows the log2 ratio of each pair of bars.

with all mutants showing similar levels of strand asymmetry (**Fig 4D**). As expected, in left (5')-replicating regions that are enriched in leading strand synthesis we observed C>A, C>T and T>G mutations predominantly. On the contrary, G>T, G>A and A>C mutations are predominantly in right (3')-replicating regions that are enriched in lagging strand synthesis.

Increased mutation density at late replicating regions: The mutation density in late-replicating regions should be higher than in early-replicating regions in MMR proficient cancer samples due to differential MMR efficiency [28]. Although all POLE mutants showed high mutational burden, they are MMR proficient with a microsatellite stable phenotype. The mutation burden of a range of mutational signatures have been associated with DNA replication timing, and a significant correlation with replication timing has been reported in cancer samples with POLE mutant associated mutational signature 10 [29]. We calculated mutation density in genomic region with distinct replication timings. As expected, all mutants similarly displayed higher mutation density in late-replicating regions than in early-replicating regions despite their different mutational context (**S7F and S7G Fig**).

Periodicity of mutation rate across and within nucleosomes: The minor groove of DNA that wraps around nucleosomes presents a differential pattern due to its physical interaction with histones, and this pattern determines periodicity in mutation rate [17,30]. Colorectal cancers with contribution from signature 10 have been reported to exhibit a positive minor-in relative increase of mutation rate as a consequence of interaction between the processes of DNA damage and repair within the nucleosome [17]. We investigated mutation rate periodicity in each specific POLE mutant separately, and we observed the positive minor-in relative increase of mutation rate in all POLE mutants to comparable levels, suggesting that the different POLE mutant induced mutations are not differentially affected by DNA-histone interactions (**S7H–S7J Fig**).

In order to investigate if the individual samples in Other-Exo group could mask any specific errors, we carried out the analysis of the samples in Other-Exo group individually as indicated in (**S8 Fig**). All mutants show very consistent mutational patterns in terms of CTCF binding sites, transcription start site, exonic and intronic regions and mutational asymmetry.

## Mutational context of POLE-mutants predisposes colorectal cancers to developing TP53 R213* mutation hotspots

Since we had identified that different POLE mutants have different mutation spectra, were interested to determine whether this may predispose cells to specific additional cancer driver mutations. We screened a list of 47 cancer driver mutation hotspots, determined based on recurrence in cohorts where we could access mutation calls in an unbiased manner (see **Methods**), in a total of 7,345 colorectal cancer samples including 47 POLE mutants (16 P286R, 15 V411L and 16 Other-Exo mutants with Sig10) and 7,298 POLE wild-type samples, to investigate if any hotspots are enriched in specific POLE mutants (**S2 Table**). For all hotspots tested, only the truncating mutation R213* in TP53 was identified to be significantly enriched in POLE P286R mutants (P = 0.0076, Fisher's exact test, Benjamini-Hochberg FDR 10%, **Fig 5A and S3 Table**). For all P286R mutants, 62.5% (10/16) harbor this hotspot, while it occurs in only 19.4% (6/31) of other POLE mutants (**Fig 5A and S3 Table**). For the remaining 7,298 POLE wild-type samples, only 2.2% (163/7298) were identified with this hotspot mutation. This nonsense mutation is a C>T transition in the context of TT[C>T]GA (**Fig 5B**), which is a relatively enriched context in P286R mutants, with adenine being more prevalent in the 5th position, compared with the other POLE mutants (**Fig 5C**, $p < 0.05$, Student's t-test, **S9 Fig**). Since the mutation also occurs at a highly methylated CpG site [20], we specifically compared the frequency of TT[C>T]GA in P286R versus V411L relative to the number of C>T

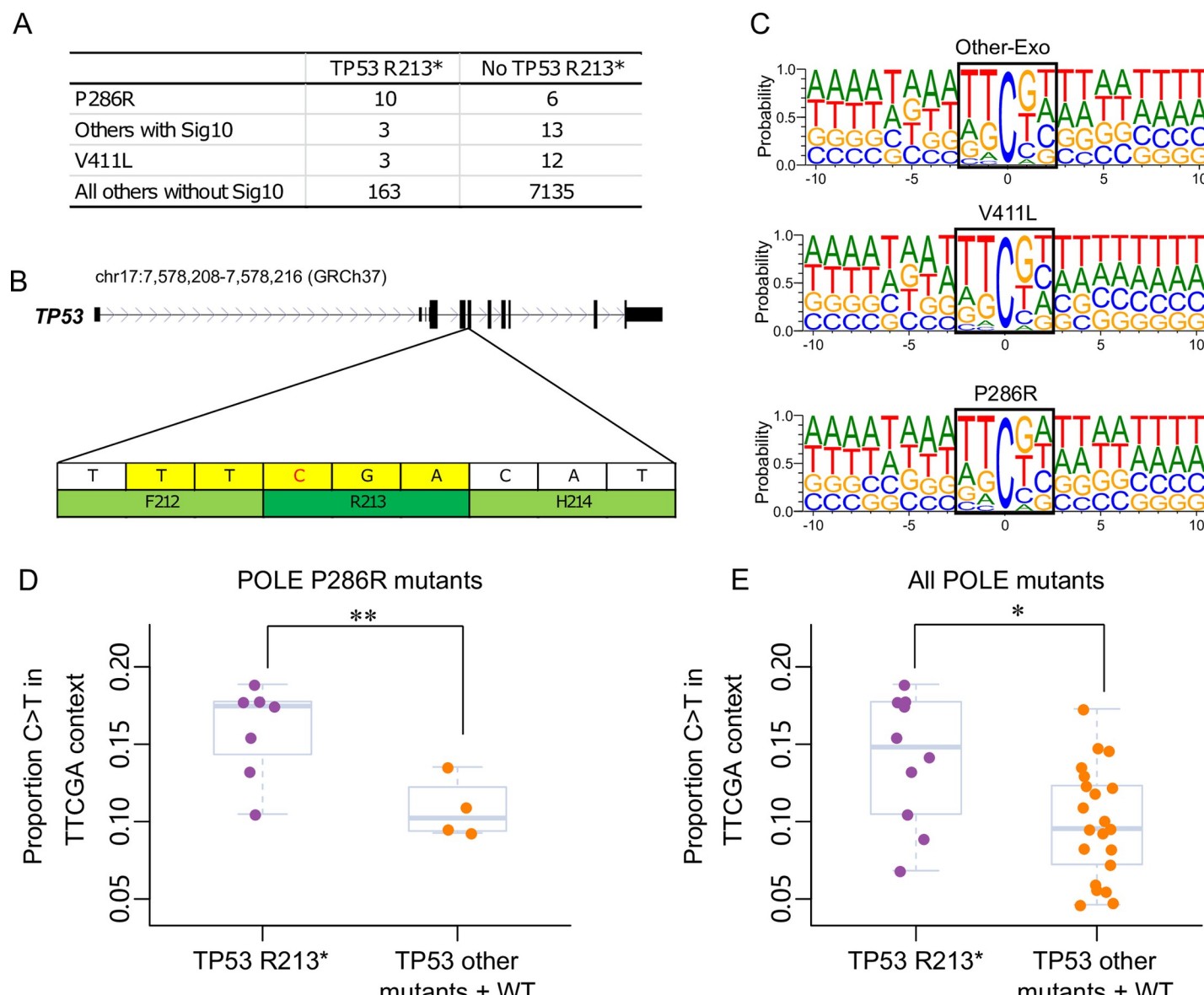

Fig 5. Mutation hotspots in POLE mutants. (A) Contingency table of different POLE-mutant and POLE wild type colorectal cancer samples with or without the TP53 R213* mutation. Samples with Sig10 were confirmed by either POLE driver mutation or mutational spectrum clustering. (B) Truncating mutation TP53 R213* was caused by C>T substitution in TT[C>T]GA context. (C) Frequency of 21-bp sequence context centered by mutated cytosine in different POLE mutants, and the penta-nucleotide contexts were indicated in black box. Proportion of TT[C>T]GA mutations in the NN[C>T]GN pentanucleotide context in all POLE P286R and V411L mutants (D) and (E) POLE P286R mutants with and without the TP53 R213* mutation. * < 0.05. ** < 0.01, Student's t-test.

mutations at CpG sites (i.e. NN[C>T]GN). It is evident that this context is significantly more enriched in POLE P286R mutants (**Fig 5D**, p < 0.01, Student's t-test). To quantify the effect this difference might have on the mutagenesis of TTCGA sites across the genome, we counted the number of sites in POLE P286R, V411L and other exonuclease domain mutants respectively and normalized the count by the total number of NN[C>T]GN mutations in each group (**S5 Table**). We find that such sites are 15% more likely to be mutated (5.68% versus 4.95% of TTCGA mutated in P286R and V411L respectively). As the relative frequency of the TT[C>T]GA pentanucleotide context differs between individual samples, we also compared its relative

frequency in POLE mutants with and without the TP53 R213* mutation. This mutational context was found to not only be significantly higher in POLE P286R mutants with the TP53 R213* mutation compared with those without this mutation (P < 0.05, Student's t-test, **Fig 5E**). This suggests that for C>T mutations at CpG sites, POLE P286R mutant colorectal cancers are more likely to form mutations at TTCGA sequence context and thus have a higher chance of acquiring the TP53 R213* mutation.

To determine if this effect is cancer specific, we also explored if the enrichment of this hotspot is present in 2045 endometrial cancer samples (**S6 Table**). TP53 R213* is not more enriched in POLE P286R mutants than other POLE mutants but it is significantly enriched in endometrial POLE mutants (P<0.001, Chi-square test, **S7 Table**) as 15.28% (11/72) POLE mutants harbor this hotpot while it is identified in 0.006% (11/1973) non-POLE mutants. This suggests that this enrichment is specific to colorectal cancer and may reflect the higher apparent positive selection for TP53 mutations in colorectal cancer compared with endometrial cancers in general with 57% (4,677/7,345) TP53 mutants in colorectal cancer versus 48% (974/2,045) TP53 mutants in endometrial cancer (**P < 0.001**, Fisher's exact test).

## Discussion

In this study, we investigated genome-wide regional mutational profiles of different POLE mutants, as well as their influence on driver mutation formation in cancer. Genomes with POLE functional defects present with differential mutation spectra but show largely similar regional mutational profiles. Significantly, we identified a recurrent nonsense mutation in TP53 that is enriched in P286R mutants, indicating a new insight into mutational processes of specific POLE mutants.

Shinbrot et al. (2014) [11] had previously characterized the functional POLE mutants with impaired exonuclease activity and describe a preference for C>A mutations in such mutants. Our study stratifies the functional POLE mutants in more detail based on the 96 trinucleotide mutational contexts and supported by expanded panel data (**S1A Fig**) and signature contribution (**S2 Fig**). We show that higher frequency of C>T mutations in V411L mutants distinguishes them from other mutants. Methylated cytosine have been shown to readily mutate to thymine as a result of methylcytosine deamination [31]. Although POLE V411L had comparably more mutations at the cytosine of CpG dinucleotides than other POLE exonuclease domain mutants (16.68% (V411L), 6.38% (P286R) and 10.19% (Other-Exo)), all mutants showed the same positive association with methylation level after adjusting for total CpG mutation count. This means that the higher level of C>T mutations in the POLE V411L compared with other POLE is not dependent on methylation, but rather sequence context is the major factor in determining the mutational spectrum of the different POLE exonuclease domain mutants. Our finding that different POLE mutants display consistent mutational distribution across genomic regions including CTCF binding sites, transcription start site, exonic and intronic regions, regions of different replication timing and regions across and within nucleosomes further confirms that the differential mutational process between POLE exonuclease mutants is principally at highly localized sequence level.

More generally, our study also highlighted some general mutagenic processes common to all POLE exonuclease domain mutants. For instance, CBSs are frequently mutated across different cancer types, and are a major mutational hotspot in noncoding cancer genomes [24]. CTCF binding sites display a specific mutation pattern in skin cancers due to differential nucleotide excision repair [18]. We observed decreased mutation density at and adjacent to CBSs, and the decline starts at around 110 nucleotide distance from the center of the CBS. It has been proposed that the decrease in mutation density in this region might be due to either

the use of an alternative polymerase [24]. CTCF-cohesin binding sites might be treated like DNA-protein crosslinks during replication, and are bypassed with the help of an accessory helicase RTEL1 and is later filled by translesion synthesis [32]. Finally, disparity in mutation rate between exon and intron regions, and early and late replication timing regions were identified in all POLE mutants, although the effect appeared strongest in the P286R mutants, possibly due to the higher mutation burden in these samples. These results suggest that mismatch repair is an important system to protect against POLE replication errors regardless of subtle differences in the way the mutations were generated.

Mutational signatures representing the spectrum of different somatic mutations can be employed to decipher the mutational process that operated within an individual cancer [33]. Recent studies have revealed the associations between mutational processes and somatic driver mutations to some extent, and indicated that altered tri-nucleotide preferences arising from a certain signature would increase the likelihood of the associated driver mutation arising [34,35]. Previous studies have identified an association between the TP53 R213* truncating mutation and POLE mutant cancers [11,20]. Our study has further identified that this TP53 hotspot is significantly enriched in POLE P286R mutants (62.5%) in colorectal cancer. The TP53 R213* truncating mutation is caused by a C>T transition in a TTCGA penta-nucleotide context, and we found that POLE mutants with this mutation do generally have higher relative frequency of this mutational context compared to POLE mutants without this mutation. This implies a possible direct causal relationship between POLE-associated mutagenesis and acquisition of this driver mutation. However, although the difference in the relative mutation frequency between POLE R286R and the other exonuclease domain mutations is significant, this difference in level is modest and thus the selection for TP53 mutations may also play a role in the final observed frequency of such mutations. Furthermore, the enrichment of TP53 R213* was not present POLE P286R mutant endometrial cancer, it is possible that selection also plays a role as TP53 mutations are more prevalent in colorectal cancers suggesting that there is stronger selective pressure for such mutations which may explain why the enrichment of the TP53 R213* mutation is only present in POLE P286R in colorectal cancer. It remains intriguing why the specific mutation spectrum varies even within cancer genomes with the same POLE mutation. Since the mutation load in POLE mutants is very high, even in targeted sequencing data, the differences are unlikely due to variations in sampling. It is known that human DNA polymerases can be postranslationally modified [36] and it may be possible that interactions with differential posttranslational regulation and POLE mutations underlies the observed differences in mutation spectra.

Finally, our work supports recent molecular and structural studies on POLE mutants. V411L and P286R are the two most frequent POLE mutants and they are located far away from each other in the exonuclease domain, thus conferring different functions [37]. Structural and molecular dynamics simulation studies in S. cerevisiae have revealed that P301R (P286R in Human) substitution prevents proper positioning of ssDNA in the exonuclease active site of Pol ε, resulting in promoting the extension of mismatched primer termini [7,8]. However, V411 is far away from exonuclease active site may function by affecting the positions of other residues adjacent to the active site [9]. In yeast, POLE mutants with a weak exonuclease activity have more C>T and less C>A mutations than mutants with no exonuclease activity [7]. We therefore speculate that the proportionally reduced C>A and increased C>T mutation loads in V411 may arise due to stronger exonuclease activity, as the mutation is distal from that site. Consistent with this, in a cell free system, V411L was found to have 3-fold reduced exonuclease activity compared to wild-type, while P286R mutants displayed a 10-fold reduction [11].

In summary, understanding how specific driver mutation may arise could lead to new targeted therapeutic strategies. This study has shown the importance of further subtyping

cancers, not only focusing on the mutated genes, but also the specific mutations within those particular genes. Stratifying samples based on DNA polymerase activity defects has enabled us to gain a better understand the mutational processes in colorectal cancer genomes.

## Methods

### Ethics statement

This study was approved by the Institutional Review Board of the University of Hong Kong/ Hospital Authority Hong Kong West Cluster (approval no. UW 18–599). All patient data analyzed in the study were acquired as anonymized data.

### Somatic mutations and sample classification

All somatic mutations of 53 whole genomes colorectal cancer were obtained from The Cancer Genome Atlas (TCGA) [38]. Microsatellite status and *POLE* mutation status were provided for each sample as listed in **S1 Table**.

2,506 colorectal samples with complete whole exome/target capture data from TCGA and previously published datasets [39–41] were first used to identify recurrent driver mutation sites (in at least 20 individuals) in colorectal cancer. Furthermore, 257 whole genome sequenced colorectal cancer samples but with only selected mutation data available [24] and another 4,582 colorectal samples also with target capture data from AACR Project GENIE through cBioPortal [42] were additionally used for analyzing POLE mutants with driver mutation hotspots. A table showing the sample cohorts and the mutation status of all samples are show in **S2 and S5 Tables**, respectively. Mutations were annotated by oncotator-1.9.9.0 when necessary.

For all samples with non-silent mutations in *POLE*, we performed clustering based on proportion of 96 tri-nucleotide context in order to distinguish functional POLE mutants that are characterized by mutational signature 10. For samples obtained from AACR Project GENIE, functional POLE mutants were confirmed by a list of reported driver mutations reported previously [2].

### Mutational signature analysis

The profile of each signature was displayed using the six substitution subtypes: C>A, C>G, C>T, T>A, T>C and T>G. For signature generated by tri-nucleotide context, each substitution was examined by incorporating information on the bases immediately 5' and 3' to each mutated base to generate 96 possible mutation types. For signature generated by penta-nucleotide context, each substitution was examined by incorporating information of two nucleotides at 5' and 3' to each mutated base resulting in 1536 possible mutation types. The mutational signatures were displayed and reported based on the observed tri-nucleotide/penta-nucleotide frequency of the human genome.

### Methylation data and mutation

Whole genome bisulfite sequencing data from normal sigmoid colon tissue were downloaded from the Roadmap Epigenomics Atlas [43]. All cytosines in the CpG di-nucleotide were merged into 12 bins according to their methylation level as: [0], (0, 0.1], .., (0.9, 1.0), [1]. These bins were then used as intersected regions to calculate mutation rate in each methylation level.

### Penta-nucleotide context normalization for each methylated level

First, the abundance of each penta-nucleotide context in which CpG context located were calculated by using the downloaded whole genome bisulfite sequencing data. Then we weighted

each context (*f*) by their counts, and made the sum of weights values equal to 1. Similarly, the abundance of penta-nucleotide context in each methylation level was also calculated and weighted (*F*). Next, the counts of mutated contexts of each sample were computed (*N*). Finally, the normalized value of each methylation level was obtained as (*C*):

$$C = \sum_{k=1}^{n} N(k) * f(k)/F(k)$$

### CTCF motifs and data analysis

CTCF/cohesin binding sites for the LoVo cell line were obtained from published paper[24]. Each CTCF motif was extended to 1000 bp on each side, and the mutation profiles were generated by counting mutations that are intersected with these sequences at each base. In order to obtain expected counts that are affected by fraction of distinct contexts, the following procedures were conducted. First, the count (*M*) of each mutated context was calculated in the overall extended sequences. Then, the abundance of each context in the whole extended sequence was computed as *A*, and for each base of each line in the stacked sequence, the relative frequency (*f*) was calculated by dividing the number of mutations with that context by the abundance *A*.

$$f = M/A$$

Next, for each position *p* within each sequence, we weighted *p* by its respective context-specific frequency *f*, and made the sum of weights across all 2001 values equal to 1. So the vector of weights $W_p$ across the specific 2001-bp sequence is given by:

$$W_p = f_p / \sum_{(k=1)}^{n} f(k)$$

Subsequently, all 2001-bp sequences were stacked and the expected count at position *p* was computed as $m^* W_p$, where *m* is the count of mutations in the sequence where *p* is located. Finally, the expected count at a given position *p* of the stack of aligned sequences is obtained as the sum of all the expected counts at each sequence of position *p*.

### Generation of mutation and profiles across transcription start sites

The information of transcription start sites (TSS) were obtained from canonical genes from the UCSC table browser. For each set of TSSs, mutation profiles were generated by counting the number of three major types of mutations (C>A, C>T and T>G) across a ±5,000-bp window centered by TSS. Mutation counts were normalized by dividing the abundance of corresponding context at each position.

### Periodicity of the relative increase of mutation rate

The methods used for mutational periodicity analysis referred to previously published paper and script [17]. Briefly, 147 bp length mid-fragments of high-coverage MNase-seq reads representing putative nucleosome dyads were obtained from the paper published by Gaffney et al [44]. Then the wig format file was converted to the bed format for following analysis. The relative positions to the dyad of two center nucleotides of the DNA to decide the minor groove facing the histones and away were obtained from the paper published by Cui and Zhurkin [45]. These positions combined with somatic mutation data were used to calculate mutation rate in stretches of DNA with the minor groove facing histones and away from them.

## Calculating mutational strand asymmetries

Replication direction was defined using replication timing profiles that are from previously published paper [14]. Left- and right-replicating regions were determined by the derivative of the profile, assigning negative and positive as left-replicating and right-replicating respectively. For a given mutation type in specific replication direction, the mutation counts ($N$) in that region were calculated, and its complementary mutation was calculated as $n$. Then, asymmetry ($A$) was calculated in a given region by:

$$A = log_2(N/n)$$

## Computing mutation density in exonic and intronic region

All gene coordinates were obtained from UCSC table browser. Each gene was divided into eight parts as 5'UTR, first exon, first intron, middle exon, middle intron, last intron, last exon and 3'UTR. As the length of sequence in each part is various, we divided every sequence into 20 bins in a given part. Sequences with length of less 20-bp were discarded. Then, the mutation density was calculated and normalized to mutations per Mb and plotted in each bin.

## Replication timing and mutation density

The replication time of different chromosome position was obtained for the HepG2 cell line from the ENCODE data portal [46]. All the sequence with known replication time was integrated into 5 bins from late to early: [-4.51712, 30.8225), [30.8225, 44.19), [44.19, 55.8262), [55.8262, 63.7717), [63.7717, 80.6964]. Mutations located in each bin were calculated.

## Supporting information

**S1 Fig. Hierarchical clustered heatmap of nine whole genome sequencing POLE mutants.** (A)Heatmap is based on cosine similarity as the value of cosine similarity is indicated in each cell. (B) Heatmap is based on COSMIC signature contribution, and each column represents one COSMIC signature.
(TIF)

**S2 Fig. Mutational spectrum of each whole genome sequencing POLE-mutant based on 96 mutational contexts, with mutation type indicated on the top panel.**
(TIF)

**S3 Fig. Hierarchical clustered heatmap of all POLE mutants based on COSMIC signature contribution.** Samples with multiple POLE mutations are underlined.
(TIF)

**S4 Fig. Mutation spectra of POLE mutants. (A)** Hierarchical clustered heatmap of the frequency of 96 types of mutational contexts for 32 POLE samples that have been whole genome, whole exome or targeted sequenced. Samples with multiple POLE mutations are underlined. **(B)** Proportion of C>T mutations in the CpA, CpC, CpG and CpT contexts. Profile of **(C)** C>A and **(D)** T>G (mutations in penta-nucleotide contexts, with genome-wide frequency of each penta-nucleotide indicated at bottom of each figure.
(TIF)

**S5 Fig. Proportion of C>T mutation in each penta-nucleotide context for each whole genome sequencing POLE-mutant, with mutation type indicated on the top panel.** (TIF)

**S6 Fig. Association of methylation and mutation in different POLE mutants in different condition.** (**A**) Correlation of methylation and mutation burden after removal of repeat sequence in CpGs. (**B**) Correlation of methylation and mutation burden in the condition of the coverage of CpGs greater than five. (**C**) Correlation of methylation and mutation burden in late replication timing CpGs. (**D**) Correlation of methylation and mutation in early replication timing CpGs. (TIF)

**S7 Fig. Mutational patterns of different genomic features.** (**A**) Somatic substitutions at CBSs with a flanking sequence of 200 bp in different POLE mutants. The expected mutation was indicated in light red color. (**B**) Profile of mutation type was showed in CBSs with a flanking 200 bp sequence. (**C**) Mutational spectrum within ± 200bp sequence centered by CBS based on 96 mutational contexts. (**D**) Mutation profile around transcription start sites in different mutants. Three primary mutation types C>A, C>T and T>G in specific context were showed, and the abundance of each context was displayed in far left panel. (**E**) Profile of mutation burden across different parts of genes in different mutants. Mutation burden was normalized by the total number of mutations in each type of mutant. Association of mutational burden and replication timing. (**F**) DNA sequence with different replication timing was divided into 5 bins, and mutational burden was calculated in each bin ordered from early-to-late. (**G**) Mutational burden was normalized by total number of mutations in each type of mutant. Periodicity of tumor mutation rate within nucleosomes in different mutants: (**H**) Other-Exo, (**I**) V411L and (**J**) P286R. For each figure, the top panel shows observed and expected mutation rate, and the bottom panel shows relative increase of mutation rate. The bottom bar is schematic representation of alternating sequences of DNA with minor groove facing toward and away from histones. (TIF)

**S8 Fig.** (A) Somatic substitutions at CBSs with a flanking sequence of 1 kilo bp in different POLE mutants. The expected mutation was indicated in light red color. (B) Mutation profile around transcription start sites in different mutants. Three primary mutation types C>A (red), C>T (blue) and T>G (green) in specific context were showed. Mutation counts were normalized by the number of corresponding context and the abundance of each context was displayed in the far left panel, together with mutation data in 100 bp bins (grey) is shown. (C) Profile of mutation burden across different part of genes in different mutants. Each part of gene was divided into 20 bins and mutation burden was calculated separately. Methylation level of each part was showed in the top panel. (D) Mutational strand asymmetry associated with replication in different mutants. Lower panel of each mutant shows the log2 ratio of each pair of bars. (TIF)

**S9 Fig. Proportion of C>T mutations in the TTCGA pentanucleotide context in POLE P286R mutants and POLE V411L mutants.** Only samples with total mutations to generate mutational contexts are included. $^*$ $< 0.05$, Student's t-test. (TIF)

**S1 Table. Sample information of 53 whole genome sequencing colorectal cancer.** (XLSX)

**S2 Table. Hotspot status across in all colorectal cancer.**
(XLSX)

**S3 Table. Significance of enrichment of each mutation hotspot in the different POLE mutants.**
(XLSX)

**S4 Table. Extended contingency table comparing enrichment of the TP53 R213* mutation in colorectal cancer.**
(XLSX)

**S5 Table. Mutation counts in TTCGA sequence context**
(XLSX)

**S6 Table. Hotspot status across in all endometrial cancer.**
(XLSX)

**S7 Table. Extended contingency table comparing enrichment of the TP53 R213* mutation in endometrial cancer.**
(XLSX)

**S8 Table. Summary of cohorts used in the study**
(XLSX)

**S9 Table. Context names of Fig 1D and Fig 3A.**
(XLSX)

**S10 Table. All mutations identified in POLE mutants.**
(XLSX)

# Acknowledgments

The authors would like to thank The Cancer Genome Atlas and other research groups for making their data available for analysis in this study.

# Author Contributions

**Conceptualization:** Jason W. H. Wong.

**Data curation:** Hu Fang, Jason W. H. Wong.

**Formal analysis:** Hu Fang, Riku Katainen.

**Funding acquisition:** Jason W. H. Wong.

**Investigation:** Hu Fang, Jayne A. Barbour, Rebecca C. Poulos, Jason W. H. Wong.

**Methodology:** Hu Fang.

**Resources:** Riku Katainen, Lauri A. Aaltonen, Jason W. H. Wong.

**Supervision:** Lauri A. Aaltonen, Jason W. H. Wong.

**Writing – original draft:** Hu Fang, Jason W. H. Wong.

**Writing – review & editing:** Hu Fang, Jayne A. Barbour, Rebecca C. Poulos, Jason W. H. Wong.

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
