## [Decision Letter · Decision Letter 0]

20 Sep 2019

Dear Dr Wong,

Thank you very much for submitting your Research Article entitled 'Mutational processes of distinct POLE exonuclease domain mutants drive an enrichment of a specific TP53 mutation in colorectal cancer' to PLOS Genetics. Your manuscript was fully evaluated at the editorial level and by three independent peer reviewers. The reviewers appreciated the attention to an important problem, but raised some substantial concerns about the current manuscript.   Reviewers question if there is sufficient novelty and field advancement potential warranting  publication of your study in PLOS Genetics.  They also raised concerns about interpretation of expected mutator features of specific POLE alleles and about robustness of apparent association between a specific POLE allele and a p53 hotspot.  There are also concerns about Supplementary data that should be added for allowing to fully evaluate your conclusions.

Based on the reviews, we will not be able to accept this version of the manuscript, but we would be willing to review again a much-revised version. We cannot, of course, promise publication at that time.

Should you decide to revise the manuscript for further consideration here, your revisions should address the specific points made by each reviewer. We will also require a detailed list of your responses to the review comments and a description of the changes you have made in the manuscript.  A version with marked up changes would also help in evaluation of your revision

If you decide to revise the manuscript for further consideration at PLOS Genetics, please aim to resubmit within the next 60 days, unless it will take extra time to address the concerns of the reviewers, in which case we would appreciate an expected resubmission date by email to plosgenetics@plos.org.

[LINK]

We are sorry that we cannot be more positive about your manuscript at this stage. Please do not hesitate to contact us if you have any concerns or questions.

Yours sincerely,

Dmitry A. Gordenin

Associate Editor

PLOS Genetics

David Kwiatkowski

Section Editor: Cancer Genetics

PLOS Genetics

Reviewer's Responses to Questions

**Comments to the Authors:**

Reviewer #1: In their paper Fang et al., describe the consequences of POLE mutations on the mutational landscape of CRC. The study curates and uses an extensive collection of samples from some 7345 samples for their analysis concluding differences between specific POLE mutations including the association of P286R with TP53 R213* mutations.

Of course, some of these analyses have already been performed as the authors volunteer (cited reference 8) but this study performs a deeper analysis. The number of samples used in the initial analysis is quite small with only 3 V411L and 3 P286R samples but the authors go on to look at mutations in and around CTCF binding sites and transcription start sites. Stand asymmetries are also explored and various other features. In the second phase of the paper the larger sample set of 47 POLE mutants and >7000 controls are used to explore the association between mutation status and specific driver alleles. The real “take home” messages of the paper are:

1. Not all POLE mutants are equal.

2. POLE can generate specific driver mutations.

3. POLE mutations are not evenly distributed around the genome. i.e. around CTCF sites.

Overall the study is well performed but some would argue predictable given some of the analyses in the original Shinbrot et al., paper, at least points 1 and 2 above, which are discussed in this paper (see Figures 3 and 6 of Shinbrot et al.,). That said I do feel that Fang et al., provides enhanced resolution and a refined analysis compared to Shinbrot et al. My only suggestion for the manuscript is that the authors provide a more critical comparison of their work to the paper of Shinbrot et al., to really bring out the novelty because at present this is not so obvious.

Reviewer #2: Review for PloS Genetics

Here the authors have explored mutation calls from 7345 colorectal cancer samples from public sources. 44 samples have POLE mutations, and for 9 of them was whole genome sequencing data available. The POLE mutant samples were divided into three different groups, P286R, V411L, and other mutations in the exonuclease domain. A strong association was found between the TP53 R213* mutation and P286R mutants. The truncation mutation occurs in a sequence context (TT[C>T]GA) that is reported to occur with a higher relative frequency in TP53 R213* samples and in most samples with a POLE P286R mutation. Furthermore, it was found that the different groups of POLE mutants exhibit distinct mutation spectra with a higher relative frequency of C>T mutations in samples with a POLE V411L mutant.

This is overall an interesting manuscript, in particular the coupling to the TP53 R213* mutation. However, there are some issues that must be discussed regarding the design of the study. In conclusion, I am currently not convinced that the results in the manuscript is a sufficient advancement to merit publication in PloS Genetics.

Major points

Figure 1A, The mutational spectra was analyzed from 53 colorectal cancer whole genomes from TCGA and four types of signatures were identified in an hierarchical clustered heatmap.

Please add Supplementary table 1 to the main manuscript, clearly showing the mutations in POLE for each of the nine samples (tumors) that was found in three groups. This is important since the study is focused on them from here on. After searching in the supplementary tables I found that 5 of the nine tumors carried more than one mutation in the exonuclease domain. How does that affect the downstream analysis? In case the mutations are located on the same allele of POLE then a second alteration anywhere in DNA polymerase epsilon can suppress or enhance the mutation rate and also affect the signature of the errors made by for instance P286R or V411L. Suppressors should be expected in case the mutation rate is very high during a period in the evolution of the cancer cells in the tumor and suppressors may affect the mutation signature.

Please specify what the Other-exo variants are when first mentioned in the text.

Grouping the Other-exo variants could mask any specific errors that could be made by individual POLE exo variants, if not show that this is not the case.

I would like to see that tumors with more than one altered amino acid in the exonuclease domain is removed from the sample set. An alternative could be to show that the mutation spectra in such a sample is identical to when only P286R or V411L is present. It may also be acceptable if the authors exclude the possibility that the POLE variants are not located on the same allele.

Are all samples in Figure 1A mismatch repair proficient? If not how are mismatch repair deficient samples divided among the four groups in in Figure 1A

Is MSS and V411L significantly different from each other in Figure 1C? If not, why are they clustered differently in Figure 1 A? I would also like to know whether P286R and Other-Exo are significantly different in figure 1C.

Lane 245, I do not understand how the mutation rate can be calculated since the number of cell divisions are unknown. Could you please explain how the mutation rate is calculated.

Lane 355-362, Could you please explain why the TP53 hotspot is significantly enriched in POLE P286R mutants, but not in POLE V411L mutants although the POLE V411L mutants have a significantly higher C>T transition frequency when compared to POLE P286R ?

Minor points

Lanes 53-55, “Previous cancer genomics studies have identified a number of mutation hotspots in POLE, however how these different POLE mutants behave in shaping the mutational landscape of cancers has not been studied.” I think this is an over-statement. Please read the review and references therein by Park and Pursell, DNA repair, title: POLE proofreading defects: Contributions to mutagenesis and cancer.

Lane 82, “Residue P286 is located in the DNA binding pocket which is adjacent to

the exonuclease active site. A change of this amino acid has been predicted to affect

the structure of the DNA binding pocket and cause polymerase hyperactivity (6).” It was earlier a prediction, but it was recently shown to be the case in a high resolution structure by Parkash et al in Nature Comm 2019. In fact, Xing et al in the companion paper show that the P to R subsititution cause the enzyme to become hyperactive and has an increased capacity to extend mismatches.

Lane 89, typo, “due to” is written twice in a row, “due to due to”

Lane 308, “P286 lies in the DNA binding pocket, which might interact with single strand DNA by directly perturbing the binding pocket.” Again, please see Parkash et al Nature Comm, 2019 that specifically show and discuss this point.

Reviewer #3: Uploaded as attachment

**Have all data underlying the figures and results presented in the manuscript been provided?**

Reviewer #1: Yes

Reviewer #2: Yes

Reviewer #3: Yes

PLOS authors have the option to publish the peer review history of their article (what does this mean?). If published, this will include your full peer review and any attached files.

Reviewer #1: No

Reviewer #2: No

Reviewer #3: No

---

## [Decision Letter · Decision Letter 1]

17 Dec 2019

Dear Dr Wong,

We are pleased to inform you that your manuscript entitled "Mutational processes of distinct POLE exonuclease domain mutants drive an enrichment of a specific TP53 mutation in colorectal cancer" has been editorially accepted for publication in PLOS Genetics. Congratulations!

Yours sincerely,

Dmitry A. Gordenin

Associate Editor

PLOS Genetics

David Kwiatkowski

Section Editor: Cancer Genetics

PLOS Genetics

Comments from the reviewers (if applicable):

Reviewer's Responses to Questions

**Comments to the Authors:**

Reviewer #1: The authors have address my questions relating to the novelty of their study and in particular made clear why their work is distinct from that of Shinbrot et al.,

Reviewer #2: The authors have answered the questions and modified the manuscript in an acceptable way.

**Have all data underlying the figures and results presented in the manuscript been provided?**

Reviewer #1: Yes

Reviewer #2: Yes

PLOS authors have the option to publish the peer review history of their article (what does this mean?). If published, this will include your full peer review and any attached files.

Reviewer #1: Yes: David Adams

Reviewer #2: No

**Data Deposition**

http://datadryad.org/submit?journalID=pgenetics&manu=PGENETICS-D-19-01384R1

**Press Queries**

---

## [Editor Report · Acceptance letter]

27 Jan 2020

PGENETICS-D-19-01384R1 

Mutational processes of distinct POLE exonuclease domain mutants drive an enrichment of a specific TP53 mutation in colorectal cancer 

Dear Dr Wong, 

We are pleased to inform you that your manuscript entitled "Mutational processes of distinct POLE exonuclease domain mutants drive an enrichment of a specific TP53 mutation in colorectal cancer" has been formally accepted for publication in PLOS Genetics! Your manuscript is now with our production department and you will be notified of the publication date in due course.

With kind regards,

Kaitlin Butler

PLOS Genetics

On behalf of:
